# Diagnostic delay of sarcoidosis: Protocol for an integrated systematic review

**Tergel Namsrai**[1], **Christine Phillips**[2], **Jane Desborough**[1] *, **Dianne Gregory**[3,4], **Elaine Kelly**[3,4], **Matthew Cook**[4], **Anne Parkinson**[1]

**1** National Centre for Epidemiology and Population Health, Australian National University, Canberra, Australia, **2** School of Medicine and Psychology, Australian National University, Canberra, Australia, **3** Sarcoidosis Australia, Australia, **4** John Curtin School of Medical Research, Australian National University, Canberra, Australia

* anne.parkinson@anu.edu.au

## Abstract

### Introduction

Sarcoidosis is a rare systemic inflammatory granulomatous disease of unknown cause. It can manifest in any organ. The incidence of sarcoidosis varies across countries, and by ethnicity and gender. Delays in the diagnosis of sarcoidosis can lead to extension of the disease and organ impairment. Diagnosis delay is attributed in part to the lack of a single diagnostic test or unified commonly used diagnostic criteria, and to the diversity of disease manifestations and symptom load. There is a paucity of evidence examining the determinants of diagnostic delay in sarcoidosis and the experiences of people with sarcoidosis related to delayed diagnosis. We aim to systematically review available evidence about diagnostic delay in sarcoidosis to elucidate the factors associated with diagnostic delay for this disease in different contexts and settings, and the consequences for people with sarcoidosis.

### Methods and analysis

A systematic search of the literature will be conducted using PubMed/Medline, Scopus, and ProQuest databases, and sources of grey literature, up to 25th of May 2022, with no limitations on publication date. We will include all study types (qualitative, quantitative, and mixed methods) except review articles, examining diagnostic delay, incorrect diagnosis, missed diagnosis or slow diagnosis of all types of sarcoidosis across all age groups. We will also examine evidence of patients' experiences associated with diagnostic delay. Only studies in English, German and Indonesian will be included. The outcomes we examine will be diagnostic delay time, patients' experiences, and factors associated with diagnostic delay in sarcoidosis. Two people will independently screen the titles and abstracts of search results, and then the remaining full-text documents against the inclusion criteria. Disagreements will be resolved with a third reviewer until consensus is reached. Selected studies will be appraised using the Mixed Methods Appraisal Tool (MMAT). A meta-analysis and subgroup analyses of quantitative data will be conducted. Meta-aggregation methods will be used to

**Data Availability Statement:** No datasets were generated or analysed during the current study. All

relevant data from this study will be made available upon study completion.

**Funding:** This review is part of the "Missed opportunities in clinical practice: Tools to enhance healthcare providers' awareness and diagnosis of rare diseases in Australia" project funded by the Commonwealth represented by Department of Health Australia (Grant ID 4-G5ZN0T7). The funders had and will have no role in study design, data collection and analysis, decision to publish or preparation of the manuscript.

**Competing interests:** The authors have declared that no competing interests exist.

analyse qualitative data. If there is insufficient data for these analyses, a narrative synthesis will be conducted.

## Discussion

This review will provide systematic and integrated evidence on the diagnostic delay, associated factors, and experiences of diagnosis delay among people with all types of sarcoidosis. This knowledge may shed light on ways to improve diagnosis delays in diagnosis across different subpopulations, and with different disease presentations.

## Ethics and dissemination

Ethical approval will not be required as no human recruitment or participation will be involved. Findings of the study will be disseminated through publications in peer-reviewed journals, conferences, and symposia.

## Trial registration

PROSPERO Registration number: CRD42022307236. URL of the PROSPERO registration: https://www.crd.york.ac.uk/PROSPEROFILES/307236_PROTOCOL_20220127.pdf.

## Background

Sarcoidosis is a rare systemic inflammatory granulomatous disease. The incidence and prevalence of sarcoidosis vary across countries, and by ethnicity and gender. In studies using national datasets or large cohorts, the reported incidence is highest in northern Europe at 11.5 per 100,000 per year in Sweden [1] and 11.3–14.8 per 100,000 per year in Denmark [2]. Lower incidences have been reported in Asian countries [3–5]. Intra-country differences attributed to race are reported in the USA where the incidence among African Americans is higher than that of other Americans [6]. In Canada, migrants accounted for 10% of cases, with South East Asian migrants disproportionately represented [7]. Higher incidence rates among women have been reported in some studies, [6, 8, 9] but not in others [10]. The pattern of disease may differ for women, and they may be diagnosed at a later age [1, 11, 12].

Sarcoidosis is a disease of unknown cause, which can manifest in any organ including heart, skin, liver, joints, nervous system and eye, but it most commonly affects the lungs [9]. Its symptoms reflect the range of organs involved, from symptoms attributable to its more common pulmonary manifestation to unusual presentations involving other organs and subtle symptoms such as fatigue and pain [13–16]. Sarcoidosis can be asymptomatic, and be discovered incidentally; in one study, 13.6% of cases of pulmonary sarcoidosis were asymptomatic [17]. There is no single diagnostic test for sarcoidosis, nor a unified, commonly used set of diagnostic criteria. Diagnosis of sarcoidosis relies on clinical manifestations along with radiological or histological evidence and exclusion of possible alternative diagnoses [18].

The reported delay of onset of sarcoidosis ranges from six months to nine years depending on the organ involvement [19, 20]. In one study in the USA, only 15.3% of cases were diagnosed at the first visit [21]. Delays in diagnosis of sarcoidosis can lead to extension of the disease, organ impairment and can be accompanied by physical suffering and exhaustion among patients.

There have been no systematic reviews of studies about determinants of delay, incorporating qualitative and quantitative research to incorporate people's experiences of the processes associated with diagnostic delay. We aim to systematically review the evidence about diagnostic delay in sarcoidosis to elucidate the factors associated with diagnostic delay for this disease in different contexts and settings, and the consequences of this delay for people with sarcoidosis. More detailed information about the factors associated with delay may help throw light on points of intervention, and strategies to ensure earlier diagnosis.

## Research objective

The aim of this integrated systematic review is to review the evidence regarding diagnostic delay in sarcoidosis. To this end, our aim is to answer two key research questions:

RQ1. What are the factors associated with diagnostic delay of sarcoidosis?

RQ2. What are patients' experience of the impact of diagnostic delay of sarcoidosis?

## Methods and analysis

### Protocol development

This study protocol has been developed in accordance with the Preferred Reporting Items for Systematic Review and Meta-Analysis Protocols (PRISMA-P) and the Cochrane Handbook for Systematic Reviews [22, 23].

### Search strategy

The search strategy was developed to ensure reproducibility and increase transparency following the PRISMA-P checklist [22]. Research questions and search terms were developed using the PICOS tool (Population/Intervention/Comparison/Outcomes/Study Design) to ensure reliability and homogeneity of search results [24]. The study is registered with PROSPERO (CRD42022289830). A systematic search of peer reviewed literature will be conducted using PubMed/Medline, Scopus, and ProQuest databases, and searches of the grey literature will include Open Access Theses and Dissertations (https://oatd.org/), ProQuest Thesis and Dissertations and the National Library of Australia. Reference lists of selected studies and review articles will also be searched.

### Search term

Search terms were developed in collaboration with research team members (TN, AP, JD, CP), and combined using Boolean operators "AND" and "OR". A preliminary exploratory search on Pubmed/Medline was conducted on 15th October 2021 (Table 1) to inform the final search strategy and determine outcomes. The preliminary search conducted during the development of the search string on Pubmed/Medline identified 162 relevant titles (Table 1). This search strategy was updated and peer reviewed (TN, AP) using the PRESS checklist [11]. The final search terms will include sarcoidosis AND ("delay in diagnosis" OR "diagnostic delay" OR "misdiagnosis" OR "time to diagnosis" OR "incorrect diagnosis" OR "missed diagnosis" OR "delayed diagnosis") without restrictions on study type, date, and language.

### Study selection (inclusion and exclusion criteria)

The eligibility of the studies identified through the literature search will be determined according to the pre-developed PICOS eligibility criteria outlined in Table 2.

**Inclusion criteria.** *Population*. Studies examining people with all types of sarcoidosis and of all ages.

**Table 1. Search string conducted on Pubmed/Medline.**

| Search number | Query | Search Details | Results |
|---|---|---|---|
| 1 | sarcoidosis [Title/Abstract] | sarcoidosis [Title/Abstract] | 26,719 titles |
| 2 | delay in diagnosis [Title/Abstract] | delay in diagnosis [Title/Abstract] | 5,899 titles |
| 3 | delayed diagnosis [Title/Abstract] | delayed diagnosis [Title/Abstract] | 8,461 titles |
| 4 | diagnostic delay [Title/Abstract] | diagnostic delay [Title/Abstract] | 3,134 titles |
| 5 | time to diagnosis [Title/Abstract] | time to diagnosis [Title/Abstract] | 2,601 titles |
| 6 | misdiagnosis [Title/Abstract] | misdiagnosis [Title/Abstract] | 16,390 titles |
| 7 | missed diagnosis [Title/Abstract] | missed diagnosis [Title/Abstract] | 2,338 titles |
| 8 | incorrect diagnosis [Title/Abstract] | incorrect diagnosis [Title/Abstract] | 1,290 titles |
| 9 | #2 OR #3 OR #4 OR #5 OR #6 OR #7 OR #8 | delay in diagnosis [Title/Abstract] OR "delayed diagnosis"[Title/Abstract] OR "diagnostic delay"[Title/Abstract] OR "time to diagnosis"[Title/Abstract] OR "misdiagnosis"[Title/Abstract] OR "missed diagnosis"[Title/Abstract] OR "incorrect diagnosis"[Title/Abstract] | 37,630 titles |
| 10 | #1 AND #9 | sarcoidosis [Title/Abstract] AND ("delay in diagnosis"[Title/Abstract] OR "delayed diagnosis"[Title/Abstract] OR "diagnostic delay"[Title/Abstract] OR "time to diagnosis"[Title/Abstract] OR "misdiagnosis"[Title/Abstract] OR "missed diagnosis"[Title/Abstract] OR "incorrect diagnosis"[Title/Abstract]) | 162 titles |

*Intervention/Exposure.* Studies about delayed diagnosis, incorrect diagnosis, missed diagnosis or slow diagnosis of sarcoidosis.

*Comparison.* Given the nature of the study there will be no comparison group.

*Outcomes.* Studies that have measures on diagnostic delay, factors of diagnostic delay, and people with sarcoidosis' experiences of diagnostic delay.

**Table 2. Inclusion and exclusion criteria.**

| PICOS | Inclusion criteria | Exclusion criteria |
|---|---|---|
| Population | Studies examining people with sarcoidosis of all ages | Animal studies |
| Intervention/ Exposure | Studies examining delayed, incorrect diagnosis, missed diagnosis or slow diagnosis of sarcoidosis | - |
| Comparison | Not applicable | - |
| Outcome | Primary outcome: diagnostic delay. | - |
| | Secondary outcomes: | |
| | i) factors of diagnostic delay | |
| | ii) people with sarcoidosis' experiences of diagnostic delay | |
| Study design | All study designs | Review articles |
| Language | English, German, Indonesian | Articles except English, German and Indonesian |
| Setting | No restriction | Clinical trials, randomized trials not reporting on delayed, incorrect diagnosis, missed diagnosis or slow diagnosis of sarcoidosis. |
| Timing | No restriction | - |

*Study design.* Qualitative, quantitative and mixed methods will be included if they meet the PICOS eligibility criteria.

*Other.* No setting or publication date limitations will be applied. Only studies in English, German and Indonesian will be included.

**Exclusion criteria.** Studies on animals, review articles, clinical trials, randomized trials not reporting on delayed diagnosis, incorrect diagnosis, missed diagnosis or slow diagnosis of sarcoidosis.

## Screening

Studies identified through the systematic search will undergo title and abstract screening followed by full text screening for those remaining.

Two review authors (TN and AP) will independently screen titles and abstracts, and full texts against the pre-developed inclusion criteria. Any conflicts will be discussed and resolved by consensus with a third reviewer (JD). Exclusion rationales will be recorded.

## Data management

To ensure an independent review and screening process of the studies identified through the literature search we will use an internet-based software, Covidence, that facilitates collaboration between reviewers while maintaining independence [25]. All search results will be imported to Covidence, and research team members will screen individually while being blinded to the choices made by other members. Any conflicts will be resolved through discussion and consensus following each screening process.

## Data extraction/Data collection

Following completion of the study selection and screening, data extraction will be conducted in four stages: 1) development of a data extraction tool, 2) peer-review of the data extraction tool, 3) piloting of the data extraction tool, and 4) final data extraction. The data extraction tool will be designed by TN based on discussions with the research team and then reviewed by the team. This will be followed by a piloting stage where two reviewers (TN and AP) will independently extract data from the same five studies and compare their results to establish consensus and validity of the data extraction tool. Final data extraction will be conducted by an individual reviewer (TN).

## Data items

The following data items will be extracted from the included studies:

1. Identification of the study

This will include name of the journal, authors, publication year, short citation, research center/university/hospital/organisation, conflict of interest, and funding/sponsorship.

2. Methods

This will include study aim, study design, diagnostic criteria used, participant demographics (mean age, sex, number of participants, ethnicity, country), recruitment process, inclusion, exclusion criteria, and statistical analysis.

3. Main findings

This will include diagnosis (sarcoidosis location), type of diagnostic delay (doctor's delay or patient's delay), diagnostic delay in months (standard deviation, standard error, confidence

interval of the diagnostic delay) factors of delay (estimates, odds ratio, relative risk, standard deviations, standard errors and confidence intervals), main symptoms, patients' experiences, and other relevant outcomes.

## Methodological evaluation/quality appraisal

The Mixed Methods Appraisal Tool (MMAT), will be used to appraise the quality of included studies (quantitative, qualitative and mixed methods) [26]. If the selected studies are only quantitative, an appropriate adapted version of Newcastle-Ottawa scale will be used depending on the study types included. The chosen quality appraisal tool will be piloted by two independent review authors (TN and AP), on a randomly selected five samples with any conflicts resolved by a third reviewer following discussion (JD). An independent reviewer (TN) will continue quality appraisal on the remaining studies.

### Assessment of risk of bias

The risk of bias of the included studies will be assessed through examining data presented as a funnel plot, a scatter plot of the effect sizes against the study sample [27]. Visual inspection of asymmetry of the funnel plot will initially be conducted, with consideration of causes, including heterogeneity, reporting bias, publication bias and chance. In cases of funnel plot asymmetry, when there are more than ten studies included in the meta-analysis, further tests for funnel plot asymmetry will be used to assess the cause of bias [28].

### Data synthesis

The extracted data items will be exported to Excel (spreadsheet software) [20]. The data synthesis will be conducted using R programming running under R studio version 4.2.1. The data synthesis process will comprise four stages: 1) data checking and cleaning, 2) data conversion, 3) descriptive statistics, 4) data examination for eligibility for inclusion in meta-analysis. In the data checking and cleaning stage, all extracted data items will be checked for misspelling and missing data will be double checked to ensure any data is not mistakenly left out. This will be followed by the conversion of outcomes to unified units (i.e., delay in diagnosis will be converted to months). In this stage, all data items will be labelled (i.e., delay in diagnosis will be labelled as a numerical value). This will be followed by summarising descriptive statistics of the included studies which will be used to examine the possibility of conducting a meta-analysis.

### Narrative synthesis and meta-analysis

A systematic narrative synthesis will be undertaken to explore the findings of included studies in relation to time from symptom onset to diagnosis, and people's experiences related to delayed diagnosis in line with guidance from the Centre for Reviews and Dissemination [29].

A meta-analysis will also be conducted if extracted quantitative data are homogenous, using a random-effects model. The main analysis will be pooled diagnostic delay in each type of sarcoidosis in conjunction with subgroup analyses (i.e., location of sarcoidosis, symptom presentation, ethnic background, health service utilisation, year of publication, country). Meta-analysis including pooling of diagnostic delay and subgroup analysis will be conducted using R studio version 4.2.1 using "meta" package. Extracted qualitative data will be meta-synthesised using meta-aggregation. Similarly, processed data (findings) from qualitative studies will be extracted and aggregated into a single set of categories, which will then be further aggregated and synthesised into a set of statements that may be useful to inform clinical practice.

### Quality of evidence

The quality/certainty of evidence for all quantitative outcomes included in a meta-analysis will be judged using the Grading of Recommendations Assessment, Development and Evaluation (GRADE) working group methodology [30]. The domains of risk of bias, consistency of effect, imprecision, indirectness, and publication bias will be used to assess the certainty of the body of evidence, which will be reported in four levels: high, moderate, low, and very low.

## Discussion

In this systematic review protocol, a detailed plan of all steps of the review including protocol development, search strategy, study selection, data extraction, quality assessment, and data synthesis has been described to ensure the production of unbiased evidence regarding the diagnostic delay of sarcoidosis, its factors, and the experiences of diagnostic delay among people with sarcoidosis. Knowledge gained from this review may throw light on ways to improve delays in diagnosis across different subpopulations, and with different disease presentations.

### Strengths and limitations

The main strength of this systematic review will be the inclusion of all study types (qualitative, quantitative, and mixed methods) to integrate and analyse the current literature about diagnostic delay, its determinants and consequences, including people's experiences of delayed diagnosis in all types of sarcoidosis. A potential limitation of the study will be insufficient studies with data on ethnic or gender differences in diagnostic delay. While there will be sufficient studies exploring pulmonary sarcoidosis, there may be insufficient studies for some rarer types to identify factors associated with delays in diagnosing sarcoidosis.

### Amendments

If the protocol is amended prior to commencing the study, these amendments (date, explanation, and rationale) will be described in the final protocol. The record will be in tabular format as recommended by the Cochrane Collaboration [23].

## Supporting information

**S1 File. Diagnostic delay of sarcoidosis: A protocol of an integrated systematic review.** (DOCX)

## Author Contributions

**Conceptualization:** Tergel Namsrai, Christine Phillips, Jane Desborough, Dianne Gregory, Elaine Kelly, Matthew Cook, Anne Parkinson.

**Data curation:** Tergel Namsrai, Jane Desborough, Anne Parkinson.

**Investigation:** Tergel Namsrai, Jane Desborough, Anne Parkinson.

**Methodology:** Tergel Namsrai, Jane Desborough, Anne Parkinson.

**Writing – original draft:** Tergel Namsrai.

**Writing – review & editing:** Christine Phillips, Jane Desborough, Dianne Gregory, Elaine Kelly, Matthew Cook, Anne Parkinson.

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
