## [Decision Letter · Decision Letter 0]

5 Aug 2022

PONE-D-22-15192Diagnostic delay of sarcoidosis: protocol for an integrated systematic reviewPLOS ONE

Dear Dr. Namsrai,

Thank you for submitting your manuscript to PLOS ONE. After careful consideration, we feel that it has merit but does not fully meet PLOS ONE’s publication criteria as it currently stands. Therefore, we invite you to submit a revised version of the manuscript that addresses the points raised during the review process.

As an aside, please note that the comment from reviewer #1 regarding duplicate publication is in error, as submitting to a preprint server and registering on PROSPERO are encouraged in PLOS ONE, although please do considering citing the medRxiv preprint for transparency. 

We look forward to receiving your revised manuscript.

Kind regards,

Consolato M. Sergi

Academic Editor

PLOS ONE

Journal Requirements:

Reviewers' comments:

Reviewer's Responses to Questions

**Comments to the Author**

1. Does the manuscript provide a valid rationale for the proposed study, with clearly identified and justified research questions?

Reviewer #1: Partly

Reviewer #2: Yes

2. Is the protocol technically sound and planned in a manner that will lead to a meaningful outcome and allow testing the stated hypotheses?

Reviewer #1: Yes

Reviewer #2: Yes

3. Is the methodology feasible and described in sufficient detail to allow the work to be replicable?

Reviewer #1: No

Reviewer #2: Yes

4. Have the authors described where all data underlying the findings will be made available when the study is complete?

Reviewer #1: No

Reviewer #2: Yes

5. Is the manuscript presented in an intelligible fashion and written in standard English?

Reviewer #1: Yes

Reviewer #2: Yes

6. Review Comments to the Author

You may also provide optional suggestions and comments to authors that they might find helpful in planning their study.

Reviewer #1: Thank you for asking me to review this protocol.

The intention is laudable in that the diagnosis of sarcoidosis is often delayed, in part due to the fact that it is uncommon, but also because the diseases manifests in multiple ways and in different organs. The systematic review does not indicate how many articles they expect to identify nor are there many in the literature to date. This reviewer finds the publication of protocols of this nature somewhat unhelpful and would prefer to see the final article as publishes in the Cochrane database or elsewhere, rather than piecemeal.

The main issues are:

The appears to be the same protocol already available on the internet, raising the issue of duplicate publication.

https://www.medrxiv.org/content/10.1101/2022.05.30.22275771v1

https://www.crd.york.ac.uk/PROSPEROFILES/307236_PROTOCOL_20220127.pdf

Is this intended to be a Cochrane review? If so, it should be listed as such and the title on their website.

Specific comments:

The choice of languages for the available articles is curious, but presumably reflects the skills of the authors- it would be useful to include other languages particularly Japanese which has a large body of literature on the topic of sarcoidosis.

The introduction is somewhat lacking in balance: it is not a rare disease (10-40 in 100,000). Seizures not are a common presentation- ophthalmological, respiratory and rheumatological symptoms are much more common.

Reviewer #2: This is expected to be a good study at conception with valid research questions and expected outcomes.

Authors have expressed their concerns on the limited number of available studies that deal with diagnostic delay in sarcoidosis. Has there been a preliminary literature search that can assure authors of being able to get reasonable number of manuscripts for review?

How do the authors plan to do statistical analysis, have they got the skill or this will be sourced out?

Assuming there are enough manuscript from literature search, I believe this study will be helpful to both patients as well as clinicians.

recommendations will be helpful in looking at causes and consequences of delay in diagnosis and way forward.

7. PLOS authors have the option to publish the peer review history of their article (what does this mean?). If published, this will include your full peer review and any attached files.

Reviewer #1: No

Reviewer #2: **Yes: **Segun Samson Odetola

---

## [Author Response · Author response to Decision Letter 0]

10 Aug 2022

If applicable, we recommend that you deposit your laboratory protocols in protocols.io to enhance the reproducibility of your results. Protocols.io assigns your protocol its own identifier (DOI) so that it can be cited independently in the future. For instructions see:

Response: We have provided a reference for the preprint as follows:

On page 8 line 120:

“The preliminary search conducted during the development of the search string on PUBMED/MEDLINE identified 162 relevant titles (Table 1).”

---

## [Decision Letter · Decision Letter 1]

14 Sep 2022

PONE-D-22-15192R1Diagnostic delay of sarcoidosis: protocol for an integrated systematic reviewPLOS ONE

Dear Dr. Namsrai,

Thank you for submitting your manuscript to PLOS ONE. After careful consideration, we feel that it has merit but does not fully meet PLOS ONE’s publication criteria as it currently stands. Therefore, we invite you to submit a revised version of the manuscript that addresses the points raised during the review process.

Please specify your study and mentioned details eligibility criteria.

Please provide screening methodology, selection methodology

Please provide exclusion criteria with details

Please provide possible limitation in the study.

The current information given were incomplete. Please provide background of study, problem statement and rational of the study.

Please provide statistical signification of the sarcoidosis disease? such as prevalence rate all around the world.

Please provide data management tools.

Please provide data collection process in detail 

Please provide data items, data synthesis methodology. All information must written in detail under separate heading.

Please give PICO in detail under each heading.

Please provide implication of study protocol and what will be benefit of this protocol.

Include some more tool to assess and compare bias in different studies. The current methodology is insufficient.

We look forward to receiving your revised manuscript.

Kind regards,

Muhammad Shahzad Aslam, Ph.D.,M.Phil., Pharm-D

Academic Editor

PLOS ONE

Additional Editor Comments:

Please specify your study and mentioned details eligibility criteria.

Please provide screening methodology, selection methodology

Please provide exclusion criteria with details

Please provide possible limitation in the study.

The current information given were incomplete. Please provide background of study, problem statement and rational of the study.

Please provide statistical signification of the sarcoidosis disease? such as prevalence rate all around the world.

Please provide data management tools.

Please provide data collection process in detail

Please provide data items, data synthesis methodology. All information must written in detail under separate heading.

Please give PICO in detail under each heading.

Please provide implication of study protocol and what will be benefit of this protocol.

Include some more tool to assess and compare bias in different studies. The current methodology is insufficient.

Reviewers' comments:

Reviewer's Responses to Questions

**Comments to the Author**

1. Does the manuscript provide a valid rationale for the proposed study, with clearly identified and justified research questions?

Reviewer #1: Yes

Reviewer #2: Yes

Reviewer #3: Yes

2. Is the protocol technically sound and planned in a manner that will lead to a meaningful outcome and allow testing the stated hypotheses?

Reviewer #1: Partly

Reviewer #2: Yes

Reviewer #3: Yes

3. Is the methodology feasible and described in sufficient detail to allow the work to be replicable?

Reviewer #1: No

Reviewer #2: Yes

Reviewer #3: Yes

4. Have the authors described where all data underlying the findings will be made available when the study is complete?

Reviewer #1: No

Reviewer #2: Yes

Reviewer #3: Yes

5. Is the manuscript presented in an intelligible fashion and written in standard English?

Reviewer #1: Yes

Reviewer #2: Yes

Reviewer #3: Yes

6. Review Comments to the Author

You may also provide optional suggestions and comments to authors that they might find helpful in planning their study.

Reviewer #1: The authors have responded to the queries raised. This reviewer would prefer to see the main findings rather than just the protocol.

Reviewer #2: Previous comments have been sufficiently addressed.

The outcome of this study is promising and will most likely fulfil objectives of the study and contributes to the body of knowledge in this field.

Reviewer #3: Thank you for the opportunity to review this protocol. The protocol is significantly improved from the previous version. I have only have a few comments.

Lines 132-135: How will the authors examine patients’ experiences associated with diagnostic delays? This is not described in detail. Will there be a survey? Will evidence be based on published literature or follow up with respective authors? In the event that there are not enough studies detailing patients’ experiences, how do the authors plan to make up for that?

Since there are no limitations on publication dates, it would be interesting to see if the study shows a trend of stagnancy or improvement in diagnostic delay over the years.

I totally agree with reviewer 1 about publishing the protocol as part of a complete systematic review and meta-analysis manuscript because that will be more informative and helpful.

7. PLOS authors have the option to publish the peer review history of their article (what does this mean?). If published, this will include your full peer review and any attached files.

Reviewer #1: No

Reviewer #2: **Yes: **Dr Segun Samson Odetola

Reviewer #3: **Yes: **Shakirat Adetunji

---

## [Author Response · Author response to Decision Letter 1]

18 Oct 2022

Editor's comments:

1. Comment 1 

Please specify your study and mentioned details eligibility criteria.

Response:

Thank you for your comment. A separate section on study selection (inclusion and exclusion criteria) has been added to provide more detail on the eligibility of the studies identified through the systematic search of the literature. 

Edit in the manuscript:

Study selection (inclusion and exclusion criteria)

The eligibility of the studies identified through the literature search will be determined according to the pre-developed PICOS eligibility criteria outlined in Table 2. 

Inclusion criteria 

Population

Studies examining people with all types of sarcoidosis and of all ages. 

Intervention/Exposure 

Studies about delayed diagnosis, incorrect diagnosis, missed diagnosis or slow diagnosis of sarcoidosis.

Comparison 

Given the nature of the study there will be no comparison group.

Outcomes

Studies that have measures on diagnostic delay, factors of diagnostic delay, and people with sarcoidosis’ experiences of diagnostic delay. 

Study design 

Qualitative, quantitative and mixed methods will be included if they meet the PICOS eligibility criteria.

Other

No setting or publication date limitations will be applied. Only studies in English, German and Indonesian will be included. 

Exclusion criteria 

Studies on animals, review articles, clinical trials, randomized trials not reporting on delayed diagnosis, incorrect diagnosis, missed diagnosis or slow diagnosis of sarcoidosis. 

2. Comment 2

Please provide screening methodology, selection methodology

Response:

Separate sections on study selection and the study screening process have been added as suggested.

Edits in the manuscript:

Study selection (inclusion and exclusion criteria) – please see changes as described above in response to comment 1

Screening 

Studies identified through the systematic search will undergo title and abstract screening followed by full text screening for those remaining.

Two review authors (** and **) will independently screen titles and abstracts, and full texts against the pre-developed inclusion criteria. Any conflicts will be discussed and resolved by consensus with a third reviewer (**). Exclusion rationales will be recorded. 

3. Comment 3

Please provide exclusion criteria with details

Response:

Thank you for highlighting an important factor. A detailed exclusion criteria section has been added to the study selection and Table 2 as well. 

Edits in the manuscript:

Exclusion criteria 

Studies on animals, review articles, clinical trials, randomized trials not reporting on delayed diagnosis, incorrect diagnosis, missed diagnosis or slow diagnosis of sarcoidosis. 

4. Comment 4

Please provide possible limitation in the study.

Response:

A separate strengths and limitations section has been added to the manuscript. 

Edits in the manuscript:

Strengths and limitations 

The main strength of this systematic review will be the inclusion of all study types (qualitative, quantitative, and mixed methods) to integrate and analyse the current literature about diagnostic delay, its determinants and consequences, including people’s experiences of delayed diagnosis in all types of sarcoidosis. A potential limitation of the study will be insufficient studies with data on ethnic or gender differences in diagnostic delay. While there will be sufficient studies exploring pulmonary sarcoidosis, there may be insufficient studies for some rarer types to identify factors associated with delays in diagnosing pulmonary sarcoidosis. 

5. Comment 5

We have expanded the introduction to include a background to the study and added a problem statement as requested. The rationale of the study has also been amended as suggested. 

Edits in the manuscript:

Background

Sarcoidosis is a rare systemic inflammatory granulomatous disease. The incidence and prevalence of sarcoidosis vary across countries, and by ethnicity and gender. In studies using national datasets or large cohorts, the reported incidence is highest in northern Europe at 11.5 per 100,000 per year in Sweden (1) and 11.3-14.8 per 100,000 per year in Denmark (2). Lower incidences have been reported in Asian countries (3-5). Intra-country differences attributed to race are reported in the USA where the incidence among African Americans is higher than that of other Americans (6). In Canada, migrants accounted for 10% of cases, with South East Asian migrants disproportionately represented (7). Higher incidence rates among women have been reported in some studies, (6, 8, 9) but not in others (10) The pattern of disease may differ for women, and they may be diagnosed at a later age (1, 11) (12).

Sarcoidosis is a disease of unknown cause, which can manifest in any organ including heart, skin, liver, joints, nervous system and eyes (13, 14), but it most commonly affects the lungs (9). Its symptoms reflect the range of organs involved, from symptoms attributable to its more common pulmonary manifestation to unusual presentations involving other organs and subtle symptoms such as fatigue and pain (15-18). Sarcoidosis can be asymptomatic, and be discovered incidentally; in one study, 13.6% of cases of pulmonary sarcoidosis were asymptomatic (19). There is no single diagnostic test for sarcoidosis, nor a unified, commonly used set of diagnostic criteria. Diagnosis of sarcoidosis relies on clinical manifestations along with radiological or histological evidence and exclusion of possible alternative diagnoses (20). 

The reported delay of onset of sarcoidosis ranges from six months to nine years depending on the organ involvement (21, 22). In one study in the USA, only 15.3% of cases were diagnosed at the first visit (23). Delays in diagnosis of sarcoidosis can lead to extension of the disease, organ impairment and can be attended by physical suffering and exhaustion among patients. 

While there are studies investigating determinants of delay, there have been few systematic reviews of these studies, and none incorporating qualitative and quantitative research to incorporate people’s experiences of the processes associated with diagnostic delay. We aim to systematically review the evidence about diagnostic delay in sarcoidosis to elucidate the factors associated with diagnostic delay for this disease in different contexts and settings, and the consequences of this delay for people with sarcoidosis. More detailed information about the factors associated with delay may help throw light on points of intervention, and strategies to ensure earlier diagnosis. 

6. Comment 6

Please provide statistical signification of the sarcoidosis disease? such as prevalence rate all around the world.

Response:

Apologies for not including this important information. We have added information about incidence rates as the study focuses on the diagnostic delay, Incidence rate is drawn from multiple countries around the world, it has been added as a range. 

Edits in the manuscript:

Sarcoidosis is a rare systemic inflammatory granulomatous disease. The incidence and prevalence of sarcoidosis vary across countries, and by ethnicity and gender. In studies using national datasets or large cohorts, the reported incidence is highest in northern Europe at 11.5 per 100,000 per year in Sweden and 11.3-14.8 per 100,000 per year in Denmark (2). Lower incidences have been reported in Asian countries (3-5). Intra-country differences attributed to race are reported in the USA where the incidence among African Americans is higher than that of other Americans (6). In Canada, migrants accounted for 10% of cases, with South East Asian migrants disproportionately represented (7). Higher incidence rates among women have been reported in some studies, (6, 8, 9) but not in others (10) The pattern of disease may differ for women, and they may be diagnosed at a later age (1, 11) (12).

7. Comment 7

Please provide data management tools.

Response:

Thank you for your comment. A separate section on data management added. 

Edits in the manuscript:

Data management 

To ensure an independent review and screening process of the studies identified through the literature search we will use an internet-based software, Covidence, that facilitates collaboration between reviewers while maintaining independence (24). All search results will be imported to Covidence, and research team members will screen individually while being blinded to the choices made by other members. Any conflicts will be resolved through discussion and consensus following each screening process. 

8. Comment 8

Please provide data collection process in detail

Response:

A detailed data extraction/collection section has been added as requested. 

Edits in the manuscript:

Data extraction/ Data collection

Following completion of the study selection and screening, data extraction will be conducted in four stages: 1) development of a data extraction tool, 2) peer-review of the data extraction tool, 3) piloting of the data extraction tool, and 4) final data extraction. The data extraction tool will be designed by ** based on discussions with the research team and then reviewed by the team. This will be followed by a piloting stage where two reviewers (** and **) will independently extract data from the same five studies and compare their results to establish consensus and validity of the data extraction tool. Final data extraction will be conducted by an individual reviewer. 

9. Comment 9

Please provide data items, data synthesis methodology. All information must written in detail under separate heading.

Response:

Thank you for pointing out this important issue. A separate and detailed section on data items and data synthesis has been added. 

Edits in the manuscript:

Data items

The following data items will be extracted from the included studies:

1. Identification of the study 

This will include name of the journal, authors, publication year, short citation, research center/university/hospital/organisation, conflict of interest, and funding/sponsorship. 

2. Methods 

This will include study aim, study design, diagnostic criteria used, participant demographics (mean age, sex, number of participants, ethnicity, country), recruitment process, inclusion, exclusion criteria, and statistical analysis. 

3. Main findings 

This will include diagnosis (sarcoidosis location), type of diagnostic delay (doctor’s delay or patient’s delay), diagnostic delay in months (standard deviation, standard error, confidence interval of the diagnostic delay) factors of delay (estimates, odds ratio, relative risk, standard deviations, standard errors and confidence intervals), main symptoms, patients’ experiences, and other relevant outcomes. 

Data synthesis

The extracted data items will be exported to Excel (spreadsheet software) (20). The data synthesis will be conducted using R programming running under R studio version 4.2.1. The data synthesis process will comprise four stages: 1) data checking and cleaning, 2) data conversion, 3) descriptive statistics, 4) data examination for eligibility for inclusion in meta-analysis. In the data checking and cleaning stage, all extracted data items will be checked for misspelling and missing data will be double checked to ensure any data is not mistakenly left out. This will be followed by the conversion of outcomes to unified units (i.e., delay in diagnosis will be converted to months). In this stage, all data items will be labelled (i.e., delay in diagnosis will be labelled as a numerical value). This will be followed by summarising descriptive statistics of the included studies which will be used to examine the possibility of conducting a meta-analysis. 

10. Comment 10

Please give PICO in detail under each heading.

Response:

A detailed section with separate headings for PICOS has been added as requested.

Edits in the manuscripts:

Study selection (inclusion and exclusion criteria)

The eligibility of the studies identified through the literature search will be determined according to the pre-developed PICOS eligibility criteria outlined in Table 2. 

Inclusion criteria 

Population

Studies examining people with all types of sarcoidosis and of all ages. 

Intervention/Exposure 

Studies about delayed diagnosis, incorrect diagnosis, missed diagnosis or slow diagnosis of sarcoidosis.

Comparison 

Given the nature of the study there will be no comparison group.

Outcomes

Studies that have measures on diagnostic delay, factors of diagnostic delay, and people with sarcoidosis’ experiences of diagnostic delay. 

Study design 

Qualitative, quantitative and mixed methods will be included if they meet the PICOS eligibility criteria.

Other

No setting or publication date limitations will be applied. Only studies in English, German and Indonesian will be included. 

11. Comment 11

Please provide implication of study protocol and what will be benefit of this protocol.

Response:

Thank you for your comment. Implications of the study protocol have been added. 

Edits in the manuscripts:

Discussion

In this systematic review protocol, a detailed plan of all steps of the review including protocol development, search strategy, study selection, data extraction, quality assessment, and data synthesis has been described to ensure the production of unbiased evidence regarding the diagnostic delay of sarcoidosis, its factors, and the experiences of diagnostic delay among people with sarcoidosis. Knowledge gained from this review may throw light on ways to improve delays in diagnosis across different subpopulations, and with different disease presentations. 

12. Comment 12

Include some more tool to assess and compare bias in different studies. The current methodology is insufficient.

Response:

Thank you for pointing out an important issue. We have included the MMAT as the methodological evaluation and NOS as an alternative choice if the included studies are all quantitative. We have also included an assessment of risk of bias as suggested. 

Edits in the manuscript:

Assessment of risk of bias

The risk of bias of the included studies will be assessed through examining data presented as a funnel plot, a scatter plot of the effect sizes against the study sample (27). Visual inspection of asymmetry of the funnel plot will initially be conducted, with consideration of causes, including heterogeneity, reporting bias, publication bias and chance. In cases of funnel plot asymmetry, when there are more than ten studies included in the meta-analysis, further tests for funnel plot asymmetry will be used to assess the cause of bias (28). 

Reviewer 1's comments:

1. Comment 1

The authors have responded to the queries raised. This reviewer would prefer to see the main findings rather than just the protocol.

Response: 

Thank you for reviewing the manuscript. The results of the systematic review will be published when completed along with the data.

Reviewer 2's comments:

1. Comment 1

Previous comments have been sufficiently addressed.

The outcome of this study is promising and will most likely fulfil objectives of the study and contributes to the body of knowledge in this field.

Response:

Thank you for reviewing the manuscript and for your supportive comments. The results of this review will be published when completed and we hope this will contribute to the body of knowledge in diagnosis of sarcoidosis. 

Reviewer 3's comments:

1. Comment 1

Lines 132-135: How will the authors examine patients’ experiences associated with diagnostic delays? This is not described in detail. Will there be a survey? Will evidence be based on published literature or follow up with respective authors? In the event that there are not enough studies detailing patients’ experiences, how do the authors plan to make up for that?

Response:

In this review we will review the available evidence of people’s experience of diagnostic delay of sarcoidosis. This evidence would most often be reported through the use of surveys or qualitative studies, including individual interviews or focus groups with people living with sarcoidosis. For this reason, we will include all types of studies (quantitative, qualitative and mixed methods).

If there is a lack of studies on people’s experience in diagnostic delay of sarcoidosis, due to the guidelines, we cannot take any additional steps as it may introduce bias to the study. However, in that case, it would mean we have identified a gap in the literature and future studies could focus on closing that gap and bringing more knowledge on the topic. 

2. Comment 2

Since there are no limitations on publication dates, it would be interesting to see if the study shows a trend of stagnancy or improvement in diagnostic delay over the years.

Response:

Thank you for the interesting point. Yes, looking at the trend would bring in good idea of the diagnostic improvement over the years. We can do an analysis on the year of publication. This was reflected in the manuscript as well. 

Edits in the manuscript:

The main analysis will be pooled diagnostic delay in each type of sarcoidosis in conjunction with subgroup analyses (i.e., types of sarcoidosis, health service utilization, year of publication).

3. Comment 3

I totally agree with reviewer 1 about publishing the protocol as part of a complete systematic review and meta-analysis manuscript because that will be more informative and helpful.

Response:

Thank you for your comment. We agree publishing the results will be important when the review is complete.

However, we are keen to publish this protocol as a methodological paper and as a foundation for our research and as a guide for future researchers.

---

## [Decision Letter · Decision Letter 2]

5 Dec 2022

Diagnostic delay of sarcoidosis: protocol for an integrated systematic review

PONE-D-22-15192R2

Dear,

We’re pleased to inform you that your manuscript has been judged scientifically suitable for publication and will be formally accepted for publication once it meets all outstanding technical requirements.

Kind regards,

Muhammad Shahzad Aslam, Ph.D.,M.Phil., Pharm-D

Academic Editor

PLOS ONE

Additional Editor Comments (optional):

Reviewers' comments:

Reviewer's Responses to Questions

**Comments to the Author**

1. Does the manuscript provide a valid rationale for the proposed study, with clearly identified and justified research questions?

Reviewer #1: Partly

Reviewer #3: Yes

2. Is the protocol technically sound and planned in a manner that will lead to a meaningful outcome and allow testing the stated hypotheses?

Reviewer #1: Yes

Reviewer #3: Yes

3. Is the methodology feasible and described in sufficient detail to allow the work to be replicable?

Reviewer #1: Yes

Reviewer #3: Yes

4. Have the authors described where all data underlying the findings will be made available when the study is complete?

Reviewer #1: No

Reviewer #3: Yes

5. Is the manuscript presented in an intelligible fashion and written in standard English?

Reviewer #1: Yes

Reviewer #3: Yes

6. Review Comments to the Author

You may also provide optional suggestions and comments to authors that they might find helpful in planning their study.

Reviewer #1: No new comments

Reviewer #3: Authors have adequately addressed my concerns. It will be interesting to see the outcome of this study and how the results bridge the gaps in knowledge of sarcoidosis diagnosis delays.

7. PLOS authors have the option to publish the peer review history of their article (what does this mean?). If published, this will include your full peer review and any attached files.

Reviewer #1: No

Reviewer #3: **Yes: **Shakirat A Adetunji

---

## [Editor Report · Acceptance letter]

7 Dec 2022

PONE-D-22-15192R2 

Diagnostic delay of sarcoidosis: protocol for an integrated systematic review 

Dear Dr. Parkinson:

I'm pleased to inform you that your manuscript has been deemed suitable for publication in PLOS ONE. Congratulations! Your manuscript is now with our production department. 

Kind regards, 

on behalf of

Dr. Muhammad Shahzad Aslam 

Academic Editor

PLOS ONE